# Addition of Vitamin C Mitigates the Loss of Antioxidant Capacity, Vitality and DNA Integrity in Cryopreserved Human Semen Samples

**DOI:** 10.3390/antiox13020247

**Published:** 2024-02-19

**Authors:** Alena J. Hungerford, Hassan W. Bakos, Robert J. Aitken

**Affiliations:** 1School of Environmental and Life Sciences, College of Engineering, Science and Environmental Science, University of Newcastle, Callaghan, NSW 2308, Australia; alena.hungerford@uon.edu.au (A.J.H.); hassan.bakos@memphasys.com (H.W.B.); 2Memphasys Ltd., Sydney, NSW 2140, Australia

**Keywords:** antioxidant, sperm freezing, oxidative stress, reductive stress, peroxidation, DNA damage

## Abstract

Cryopreservation of human spermatozoa is a necessity for males suffering from infertility who cannot produce fresh semen for insemination. However, current ART cryopreservation protocols are associated with losses of sperm motility, vitality and DNA integrity, which are thought to be linked to the induction of oxidative damage and the toxic properties of commercial cryoprotectants (CPAs). Preventing or mitigating these losses would be hugely beneficial to sperm survival during ART. Therefore, in this in vitro investigation, lipid peroxidation, production of reactive oxygen species, movement characteristics, antioxidant capacity, vitality, and DNA integrity were examined in semen samples both pre- and post-cryopreservation with CPA supplementation. The findings revealed a 50% reduction in antioxidant capacity with CPA addition, which was accompanied by significant increases in generation of reactive oxygen species and formation of lipid aldehydes. These changes were, in turn, correlated with reductions in sperm viability, motility and DNA integrity. Antioxidant supplementation generated bell-shaped dose-response curves with both resveratrol and vitamin C, emphasising the vulnerability of these cells to both oxidative and reductive stress. At the optimal dose, vitamin C was able to significantly enhance vitality and reduce DNA damage recorded in cryopreserved human spermatozoa. An improvement in sperm motility did not reach statistical significance, possibly because additional pathophysiological mechanisms limit the potential effectiveness of antioxidants in rescuing this aspect of sperm function. The vulnerability of human spermatozoa to reductive stress and the complex nature of sperm cryoinjury will present major challenges in creating the next generation of cryoprotective media.

## 1. Introduction

As levels of environmental pollution and psychological stress and rates of delayed childbearing have increased globally, the use of artificial reproductive technologies (ART), especially in vitro fertilisation (IVF) and intra-cytoplasmic sperm injection (ICSI), have risen in parallel [1,2]. Currently one in ten to one in four couples experience difficulties conceiving, with 35–50% experiencing this difficulty due to a male factor [3,4,5]. A therapeutic strategy that comprises an important component of ART is the cryopreservation of human spermatozoa—cooling the cells to −196 °C followed by storage in liquid nitrogen Dewars, allowing for sperm banking prior to insemination [6,7]. When the spermatozoa are stored for use at a later date, the reproductive abilities of males can be saved from the harmful effects of spermatogenesis-compromising cancer treatments or ageing. Fertility can also be preserved following vasectomy or for men in dangerous occupations such as the military, and semen samples that cannot be obtained on the day of oocyte retrieval can still be utilised [8,9].

Despite ongoing research into cryogenics, there has been a lack of significant improvement in the quality of human spermatozoa following cryopreservation, with the addition of cryoprotectant and the cryopreservation process itself leading to significant reductions in sperm vitality, motility and DNA integrity [10,11,12,13,14,15]. Spermatozoa are particularly vulnerable to cryoinjury because of the lipid composition of their plasma membranes and their limited capacity for active repair [11]. In this context, it has become apparent that spermatozoa suffer from many different kinds of disruption as a consequence of cryopreservation, including osmotic stress, oxidative stress, and the physical damage induced by ice-crystal formation [14,16]. Given the importance of these pathophysiological mechanisms, numerous attempts have been made to improve cryopreservation outcomes by incorporating a variety of reagents into the cryodiluent to counteract these stressors. Such modifications have included (i) the incorporation of antioxidants such as L-carnitine, vitamin C, N-acetyl-cysteine, resveratrol, silymarin, and melatonin to limit the creation or impact of reactive oxygen species (ROS); (ii) the addition of cell-permeant reagents to suppress ice-crystal formation, including dimethyl sulfoxide, glycerol, and/or ethylene glycol; and (iii) the addition of membrane-impermeant sugars (trehalose) and esters (crocin, a crocetin di-gentiobiose ester) that can help control both oxidative and osmotic stress [17,18,19,20,21,22].

The development of an optimised cryoprotective agent (CPA) capable of reducing oxidative stress and increasing levels of motility, vitality and DNA integrity, could promote high fertilisation rates, reduce embryonic mortality, and increase embryo implantation [21,22,23,24,25]. Such improvements might in turn support the use of cryopreservation for severely compromised oligospermic males whose spermatozoa might not otherwise have survived the procedure [1]. The aim of the research reported in this paper was to comprehensively analyse the relationships between antioxidant availability, oxidative stress and sperm quality and then to use this information to develop improved conditions for the low-temperature storage of human spermatozoa.

## 2. Materials and Methods

### 2.1. Materials

Unless otherwise stated, all reagents were obtained from Sigma-Aldrich (Burlington, MA, USA).

### 2.2. Semen Collection and Analysis

This study used human spermatozoa from the University of Newcastle reproductive research program, comprising unselected healthy donors of unknown fertility status, aged 18–75. The mean ± SE parameters of semen quality in this cohort of normozoospermic donors were as follows: sperm count (66.5 ± 6.2 × 10^6^/mL), vitality (77.8 ± 1.2%), motility (56.0 ± 2.0%), progressive motility (22.3 ± 1.5%), and morphology (6.1 ± 0.2%). Ethical approval from the University of Newcastle human ethics committee and the state government was secured for the use of these semen samples in research (H-2013-0319 and 200621). After at least 48 h abstinence, semen samples were produced by masturbation and collected into sterile sample containers, which were delivered to the laboratory within 1 h of ejaculation. Participants were informed that their samples would be used for research purposes, and signed consent was given. Sperm vitality was assessed by light microscopy at ×400 magnification using the eosin method and counting 200 cells [26]. Sperm morphology was measured by the examination of spermatozoa placed on pre-stained slides (Testsimplets Waldeck, GmbH, Münster, Germany) under oil immersion and scored at ×1000 magnification. A total of 200 cells were counted to determine the percentage of morphologically normal cells. Sperm counts were assessed using a Neubauer haemocytometer, as described by the World Health Organisation (2021).

### 2.3. Evaluation of Sperm Motility

Sperm motility and movement characteristics were assessed using a CASA system [Hamilton Thorne, IVOS II, Beverly, MA, USA). The CASA settings for this analysis were as follows: minimum total count 200; kinematic settings: progressive STR (%) 80, progressive VAP (µm/s) 25, slow VAP (µm/s) 5, slow VSL (µm/s) 5, static VAP (µm/s) 0, static VSL (µm/s) 0; objective Zeiss 10× NH IVOS-II 160 mm.

### 2.4. Cryopreservation and Thawing of Samples

Cryopreservation was achieved by slowly mixing an aliquot of semen in a 1:1 ratio (*v*/*v*) with Quinn’s Advantage^TM^ Sperm Freezing Medium, a commercial HEPES-buffered salt solution containing human serum albumin, glycerol, sucrose, and gentamicin (Cooper Surgical, Shelton, CT, USA), then loaded into 0.5 mL straws (CBS High Security Sperm Straw with Tripartite Plug, IMV, L’Aigle, France: Conception Technology, San Diego, CA, USA) with the ends heat-sealed. The straws were gradually cooled in liquid nitrogen by controlled freezing in the FreezeControl^TM^ system from Cryologic (Victoria, Australia) on setting 0. After freezing to −80 °C, the straws were placed in goblets and stored at −196 °C in a liquid nitrogen Dewar for a maximum of 4 months [15].

Thawing was conducted by removing the straws from the Dewar and allowing them to acclimatise to room temperature in the laboratory over 15 min. After acclimatisation, the straw exteriors were washed with distilled water and dried, and then the straw ends were cut and the contents released into 1.5 mL Eppendorf tubes. Following thawing, the samples were processed within the next 15 min to avoid unnecessary damage from continued exposure to the cryoprotectant.

### 2.5. Sample Preparation

#### 2.5.1. Overall Impact of CPA

First, 1.4 mL of samples (*n* = 15) had a 200 µL aliquot removed for use as a control, while the remainder was diluted 1:1 with cryoprotectant that had been removed from the fridge, brought to room temperature (Quinn’s Advantage^TM^ Sperm Freezing Medium, CooperSurgical, Trumbull, CT, USA) and incubated for 5–8 min. A 200 µL aliquot was removed and centrifuged at 500× *g* for 3 min, and the spermatozoa were resuspended in 200 µL Biggers, Whitten and Whittingham (BWW) medium supplemented with 1 mg/mL polyvinyl alcohol (PVA) for further analysis [27]. The remaining solution was cryopreserved and stored at −196 °C in a liquid nitrogen Dewar. Upon thawing, the solution was centrifuged at 500× *g* for 3 min and the spermatozoa were resuspended in BWW and analysed both immediately and following incubation in the dark for 3 h at room temperature (24 °C) or 37 °C, as these temperatures are both utilised in ART settings. The elements of sperm function examined were sperm motility (CASA), vitality, mitochondrial ROS generation (MSR), cellular ROS generation (DHE), lipid aldehyde formation (4-HNE and MDA) and level of DNA damage (HALO). In addition, antioxidant capacity (free radical and hydrogen peroxide scavenging activities) was measured in the original semen and in the semen diluted with Quinn’s cryoprotectant medium.

#### 2.5.2. Analysis of the Protective Impact of Resveratrol

First, 1 mL of sample (*n* = 15) had a 50 µL aliquot removed for use as a control; 190 µL was then removed and mixed with 10 µL of 100% ethanol as the vehicle control, and the remainder was diluted 1:1 with either warmed cryoprotectant or warmed cryoprotectant containing 10, 20, 40, 80, 160, and 320 µM resveratrol. The elements of sperm function examined were sperm motility (CASA), vitality, and antioxidant capacity (free radical scavenging activity), all measured in the semen/cryopreservative mixture. Examinations were conducted 5–8 min post-dilution to mitigate the harmful effects of glycerol exposure with time and again following cryopreservation.

#### 2.5.3. Impact of the Addition of Three Water-Soluble Antioxidants 

First, 0.5 mL of individual samples (*n* = 15) had a 50 µL aliquot removed for use as a control, while the remainder was diluted 1:1 with either warmed cryoprotectant or warmed cryoprotectant supplemented with (i) 0.4 mM Vitamin C (L-ascorbic acid), (ii) 2 mM melatonin, or (iii) 0.2 mM N-acetyl cysteine (NAC). These doses were determined on the basis of dose-response analyses conducted in house to determine which dose of each compound would restore the antioxidant protection provided to the spermatozoa to the original value. The elements of sperm function examined were sperm motility (CASA), vitality, and antioxidant capacity (free radical scavenging activity), all measured in the neat semen and following the addition of Quinn’s cryoprotectant. Examinations were conducted within 8 min post-dilution to mitigate the harmful effects of glycerol exposure with time.

#### 2.5.4. Dose-Dependent Study of Vitamin C Addition Pre- and Post-Cryopreservation

First 1.7 mL of individual samples (*n* = 15) had a 100 µL aliquot removed to serve as a non-cryopreserved control; then, 1 mL was aliquoted out and diluted 1:1 with either warmed cryoprotectant or warmed cryoprotectant supplemented with 0.1, 0.2, 0.4, 0.8, and 1.6 mM vitamin C for 5–8 min before cryopreservation and storage at −196 °C in a liquid nitrogen Dewar. Upon thawing, the solution was centrifuged at 500× *g* for 3 min and the spermatozoa were resuspended in BWW and analysed. The elements of sperm function examined were sperm motility (CASA), vitality, and level of DNA damage (HALO).

The remaining 0.6 mL of sample was diluted 1:1 with either warmed cryoprotectant or warmed cryoprotectant with 0.1, 0.2, 0.4, 0.8, and 1.6 mM vitamin C addition. Examinations were conducted within 5–8 min post dilution to mitigate the harmful effects of glycerol exposure and included sperm motility (CASA), vitality, and antioxidant capacity (free radical scavenging activity), all measured in the semen and cryoprotectant mixture.

### 2.6. Analytic Procedures

#### 2.6.1. Flow Cytometric Analysis

All flow cytometry was performed using a FACSCanto II flow cytometer (Becton Dickinson, Franklin Lakes, NJ, USA) with a 488-nm solid state laser. Emission measurements were taken using 530/30 nm band pass (green/FITC), 585/42 nm band pass (red/PE), >670 nm long pass (far red/PerCp) and 780/60 nm band pass (far red/PECy7) filters. Using the forward scatter/side scatter dot plot, a gate was drawn and placed around the sperm population, excluding debris from the analysis. A minimum of 10,000 cells were analysed from each sample using FACSDiva V8.01 software (Becton Dickinson, CA, USA).

#### 2.6.2. Flow Cytometric Measurement of Cellular 4-Hydroxynonenal Adduction

The levels of a lipid aldehyde 4-hydroxynonenal (4-HNE) were quantified to determine the rate of lipid peroxidation in the spermatozoa. Spermatozoa were incubated with LIVE/DEAD™ Fixable Far Red Dead Cell Stain (Molecular Probes, Eugene, OR, USA) at a 1:1000 dilution in BWW for 15 min at 37 °C; incubation was followed by a single centrifugation and resuspension in BWW. The sample was then incubated with a primary anti-4HNE antibody (Molecular Probes, Eugene, OR, USA) at a 1:50 dilution for 30 min at 37 °C. After the second incubation and subsequent wash, secondary antibody (AlexaFluor-488 goat anti-rabbit IgG, Molecular Probes, Eugene, OR, USA) was added at a 1:100 dilution and incubated at 37 °C for 10 min. The samples were then washed twice by centrifugation and resuspended in BWW. A snap-frozen sperm sample was used as a positive LIVE/DEAD control. Immunocytochemical analysis of the pattern of 4-HNE adduction was also performed using a Zeiss Axio A.2 fluorescence microscope (Carl Zeiss AG, Jena, Germany).

#### 2.6.3. Flow Cytometric Measurement of Malondialdehyde

The lipid aldehyde malondialdehyde (MDA) was also quantified to determine the rate of lipid peroxidation in the spermatozoa. Spermatozoa were incubated with LIVE/DEAD™ Fixable Far Red Dead Cell Stain (Molecular Probes, Eugene, OR, USA) at a 1:1000 dilution for 15 min at 37 °C, followed by a single centrifugation and resuspension in BWW. The sample was then incubated with a primary anti-MDA antibody (Abcam) at a 1:100 dilution for 30 min at 37 °C. After the second incubation and subsequent wash, secondary antibody (AlexaFluor-488 goat anti-rabbit IgG, Molecular Probes, Eugene, OR, USA) was added at a 1:100 dilution and incubated at 37 °C for 10 min. The samples were then washed twice by centrifugation and resuspended in BWW. A snap-frozen sperm sample was used as a positive LIVE/DEAD control. Immunocytochemical analysis of the pattern of MDA adduction was also performed using a Zeiss Axio A.2 fluorescence microscope (Carl Zeiss AG, Jena, Germany).

#### 2.6.4. Flow Cytometric Measurement of Mitochondrial ROS

MitoSOX™ Red dye (MSR; Molecular Probes, Eugene, OR, USA) was used to determine levels of mitochondrial ROS in the spermatozoa. Spermatozoa were incubated with 2 μM mitoSOX Red (MSR; Molecular Probes, Eugene, OR, USA) and 5 nM Sytox Green (SyG) vitality stain (Molecular Probes, Eugene, OR, USA) for 15 min at 37 °C. That step was followed by a single centrifugation and resuspension in BWW. The samples were assessed via flow cytometry, with the results expressed as the percentage of live cells that are MSR positive. As a positive control, a sperm sample was treated with 50 μM arachidonic acid (AA), while a snap-frozen sperm sample was used as a positive SyG control [26].

#### 2.6.5. Flow Cytometric Measurement of Cellular ROS

Total cellular ROS generation was measured using dihydroethidium (DHE; Molecular Probes, Eugene, OR, USA). Spermatozoa were incubated with 2 μM dihydroethidium (DHE; Molecular Probes, Eugene, OR, USA) and 5 nM SyG vitality stain (Molecular Probes, Eugene, OR, USA) for 15 min at 37 °C. This step was followed by a single centrifugation and resuspension in BWW. The samples were assessed via flow cytometry, and the results were expressed as the percentage of live cells that were DHE positive. As a positive control, a sperm sample was treated with 50 μM arachidonic acid (AA), while a snap-frozen sperm sample was used as a positive SyG control [26].

#### 2.6.6. DNA Integrity: Sperm Chromatin Dispersion (HALO)

The sperm chromatin-dispersion assay (HALO) was performed to determine the rates of DNA fragmentation in spermatozoa. Spermatozoa were snap-frozen and, once thawed, mixed with 1% low-melting agarose to a final concentration of 0.7% agarose. Then, 60 µL was spread onto precoated 0.65% agarose slides with a glass coverslip. Samples were allowed to set at 4 °C for 4 min before the coverslips were removed. Sitting horizontally, the slides were then covered in 0.08 M HCl for 7 min, gently drained, and covered in 100 mM DTT in Tris buffer 1 (4.84 g of Tris, 10 mL of 10% SDS, 10 mL of 0.5 M EDTA made up to 100 mL with MilliQ; pH 7.5) for 10 min. Following gentle draining, the slides were then covered in Tris buffer 2 (4.84 g of Tris, 11.69 g of NaCl, 10 mL of 10% SDS made up to 100 mL with MilliQ; pH 7.5) for 5 min. The samples were gently drained again and covered in Tris-Boric Acid-EDTA Buffer (TBE: 5.4 g of Tris, 2.75 g of boric acid, 2 mL of 0.5 M EDTA made up to 100 mL with MilliQ; pH 7.5) for 2 min. Once drained, the slides were immersed in increasing strengths of ethanol (70%, 90% and 100% ethanol) for 2 min at each step. The slides were finally air-dried and subsequently stained with 4′,6-diamidine-2′-phenylindole dihydrochloride (DAPI, Sigma-Aldrich, Burlington, MA, USA) solution at a 1:2000 dilution in phosphate-buffered saline (PBS: pH 7.4, 137 mM NaCl, 2.7 mM KCl, 8 mM Na_2_HPO_4_, and 2 mM KH_2_PO_4_) for 10 min. The slides were rinsed with PBS, then 30 µL of Mowiol^®^ was added and coverslips were applied. Spermatozoa were then assessed under a fluorescence microscope at ×400 magnification. A ‘halo’ of fluorescent DNA indicated that the DNA was undamaged, while the absence of a ‘halo’ indicated pre-existing single-strand breaks. For scoring purposes, the cells were classified into one of 5 categories: large halo, medium halo, small halo, no halo, or degraded spermatozoa. The percentage of DNA-damaged spermatozoa was given by the percentage of cells falling into the small halo, no halo and degraded categories.

#### 2.6.7. Antioxidant Capacity

Two types of antioxidant activity were measured using ABTS (2,2′-azino-bis(3-ethylbenzothiazoline-6-sulfonic acid) as the probe. The first of these tests examined the ability of a given sample to suppress hydrogen peroxide-induced formation of the ABTS^+•^ radical cation [28,29]. For this reaction, the buffer employed was a 50 mM phosphate buffer at pH 6.5 (1.197 g sodium phosphate dibasic heptahydrate and 1.109 g sodium phosphate monobasic monohydrate in 250 mL Milli Q water). The final reaction mixture contained 735 µL buffer, 15 µL ABTS (150 µM), 100 µL diluted sample (0.15% semen or semen/cryoprotectant mixture), and 50 µL HRP (0.05 mg/mL). The reaction was activated by the addition of 100 µL hydrogen peroxide (30 µM) and incubated at room temperature for 10 min to allow formation of the coloured ABTS^+•^ radical cation. The absorbance was finally read at 734 nm in a plate reader (SPECTROstar Nano, BMG Labtech). Controls for sample turbidity were created by omitting ABTS, while a no-sample control was created by adding distilled water instead of diluted semen. The level of hydrogen peroxide-scavenging activity was calculated as follows:[(No-Sample control) − (Sample)] + [(Sample Turbidity) − (No-Sample Control Turbidity)]

Trolox was used to calibrate this assay, and the results were expressed in Trolox equivalents. Standards were created from a 5 mM Trolox stock solution so that the final concentrations ranged from 0 to 20 µM.

The ability of semen samples to reduce the preformed ABTS^+•^ radical cation was also analysed [30]. This assay was conducted in 50 mM phosphate buffer at pH 4.8 (20.1 mg sodium phosphate dibasic heptahydrate, 1.71 g sodium dihydrogen phosphate monohydrate, and 4.5 g sodium chloride in 500 mL MilliQ water). After an initial reading (T0) at 734 nm, 15 µL aliquots of sample were added to each cuvette, mixed, and allowed to stand for 5 min at room temperature. At the end of this period, the absorbance was measured again. The corrected absorbance was then calculated as follows:[(T0Anode) − (T5Anode)] + [(T5Cathode) − (T0Cathode)]

Standards were created from a 5 mM Trolox stock solution (6.26 mg in 5 mL of 50% ethanol) so that the final concentrations following addition of 15 µL of the standard to 1 mL of activated ABTS+ ranged from 0 to 30 µM.

### 2.7. Statistical Analysis

All data were analysed by JMP, with linear regression analysis, ANOVA, and comparisons between group means using the Tukey–Kramer HSD post hoc test. Where necessary, the normality of the data distribution was improved using log or square-root transformation, as indicated.

## 3. Results

### 3.1. Changes in Antioxidant Protection following Addition of CPA

An analysis of the antioxidant protection provided to spermatozoa following addition of a commercial cryopreservation medium that is in widespread clinical use (Quinn’s Advantage Sperm Freeze) revealed that the medium possesses very little antioxidant activity, as expressed in terms of its ability to scavenge both free radicals and hydrogen peroxide (Figure 1A,B). As a result, when this medium was added to human semen in the recommended ratio of 1:1, it effectively halved the level of antioxidant protection provided to the spermatozoa (*p* < 0.001). In contrast, subsequent cryostorage of the sample had no significant impact on the levels of antioxidant activity detected (Figure 1A,B).

In order to determine whether such a dramatic change in antioxidant protection impacted the biological functionality of the spermatozoa, the movement characteristics of the spermatozoa were investigated using a CASA system. This analysis revealed that the simple addition of Quinn’s medium to human semen resulted in a rapid suppression of sperm motility (*p* < 0.001) that was decreased further still by the cryopreservation process itself (*p* < 0.001). Post-thaw, the motility of the spermatozoa remained unchanged for 3 h, regardless of whether the cells were incubated at room temperature (~22 °C) or 37 °C (Figure 2A). Progressive motility also declined following the addition of Quinn’s medium (*p* < 0.5) and again following cryopreservation (*p* < 0.05) (Figure 2B). While cryopreservation impacted percentage motility, there was no change in sperm velocity measurements (Figure 2C–E). However, there was a significant increase in beat cross frequency (BCF) (*p* < 0.05) post-addition of Quinn’s, and this value remained elevated following incubation (Figure 2F).

The addition of Quinn’s medium did not impact the vitality of the spermatozoa or the capacity of the live cells to generate ROS. However, following cryopreservation, a significant negative impact on sperm vitality (*p* < 0.01) was noted in concert with a significant increase in mitochondrial ROS generation (*p* < 0.05). Moreover, the increase in mitochondrial ROS generation was particularly marked if the spermatozoa were incubated for 3 h at 37 °C following a freeze–thaw cycle (*p* < 0.001) (Figure 3A,B).

Cellular ROS generation displayed a similar pattern, reaching statistical significance following a 3 h incubation at 37 °C (*p* < 0.05, Figure 3C). Across the entire dataset, the percentage of the live population that were positive for MSR and DHE were highly correlated (*p* < 0.001; R^2^ = 0.5; Figure 3D), suggesting that most of the ROS generated by these cells originate in the mitochondria. Some samples exhibited particularly high rates of mitochondrial ROS generation (Figure 3D, circled).

Consistent with the increase in ROS generation observed following cryopreservation, this process was associated with highly significant overall changes (*p* < 0.001) in the lipid peroxidation status of the spermatozoa, as reflected in the percentage of cells positive for the lipid aldehyde MDA (Figure 4A). Thus, while the addition of Quinn’s medium did not induce significant lipid peroxidation, this form of damage increased following cryostorage and became statistically significant, particularly following a 3 h incubation at room temperature or 37 °C (*p* < 0.001).

Across the entire dataset, the percentage of MDA-positive cells was significantly negatively correlated with total motility (*p* < 0.001; R^2^ = 0.24) and less strongly correlated with progressive motility (*p* < 0.001; R^2^ = 0.15) (Figure 4B,C). This pattern was also observed for another aldehyde marker of lipid peroxidation, 4-HNE. Again, addition of Quinn’s medium did not enhance 4-HNE formation per se; however, the formation of this aldehyde was significantly enhanced by a freeze–thaw cycle (*p* < 0.05; Figure 4D). As might be anticipated, the expression of 4-HNE was also negatively correlated with both total and progressive sperm motility (*p* < 0.01; R^2^ = 0.1), as well as with vitality (*p* < 0.001; R^2^ = 0.2; Figure 4E). Moreover, across the entire data set, 4-HNE and MDA expression were significantly correlated with each other (*p* < 0.001; R^2^ = 0.32; Figure 4F).

The correlation between MDA and 4-HNE was also supported by a microscopic examination of the subcellular patterns of aldehyde adduction in human spermatozoa (Figure 5). The images presented in this figure demonstrate that common areas of susceptibility to aldehyde adduction were revealed when the samples were probed with antibodies against each of these products of lipid peroxidation. Thus, the midpiece containing the sperm mitochondria is preferentially adducted by both 4-HNE and MDA. Both aldehydes also bound to the acrosomal domain, although this effect was much stronger with 4-HNE than with MDA. Weak punctate labelling was also evident along the length of the flagellum with both probes, possibly explaining the impact of lipid peroxidation on sperm movement [31].

Similar to the oxidative-stress parameters, chromatin integrity, as measured by the HALO assay, revealed that while the addition of Quinn’s did not induce an increase in sperm DNA fragmentation, the act of cryopreservation induced a significant increase in DNA damage (*p* < 0.05) that was further exacerbated by a 3 h incubation at room temperature or 37 °C (*p* < 0.001; Figure 6A). The degree of DNA fragmentation was negatively correlated with motility (*p* < 0.001; R^2^ = 0.27) and progressive motility (*p* < 0.001; R^2^ = 0.28) but positively correlated with free-radical generation, as measured by DHE (*p* < 0.001; R^2^ = 0.28) and MSR (*p* < 0.001; R^2^ = 0.18) (Figure 6B,C).

### 3.2. Analysis of the Protective Impact of Resveratrol

As the decrease in motility and increase in DNA damage were related to the induction of oxidative stress and a concomitant reduction in the antioxidant protection provided by the extracellular fluids, the potential benefits of adding antioxidants to the cryostorage medium was examined next. The first antioxidant chosen to compensate for the negative impact of Quinn’s medium on seminal antioxidant activity was resveratrol. A dose-response study demonstrated that the addition of resveratrol did, as anticipated, significantly increase the free radical scavenging capacity of semen extended with Quinn’s cryopreservation medium (*p* < 0.001) and that doses ≥40 µM were required to adjust seminal antioxidant capacity so that it was not significantly different from the levels recorded in the original semen samples (Figure 7A). Analysis of sperm motility under these circumstances demonstrated while the addition of Quinn’s cryopreservation medium significantly decreased sperm motility (*p* < 0.05), the addition of resveratrol ameliorated this situation such that at doses of 10 and 20 µM, motility was not significantly different from the value recorded in the original semen sample. At higher doses of resveratrol, an inhibitory impact on sperm motility was observed (Figure 7B). Resveratrol also counteracted the inhibitory impact of Quinn’s cryostorage medium on sperm vitality at a dose of 20 µM, but, again, this effect was reversed at higher doses of antioxidant (Figure 7C). With cryostorage, resveratrol did not significantly improve sperm motility at any of the doses tested.

### 3.3. Impact of the Addition of Three Water-Soluble Antioxidants 

To circumvent resveratrol’s relative hydrophobicity and requirement for an additional solvent (ethanol), a preliminary trial was conducted with three water-soluble antioxidants (vitamin C, melatonin, and N-acetyl cysteine) at doses calculated to replace the loss of antioxidant activity seen upon the addition of Quinn’s medium. All three antioxidants improved antioxidant activity to the point that the levels recorded were not significantly different from those observed in the initial semen sample (Figure 8A). The loss of vitality induced by the addition of Quinn’s medium could be prevented by the addition of vitamin C and melatonin but not by the addition of NAC (Figure 8B). None of these antioxidants significantly improved sperm motility, whether this analysis was performed before or after cryo-preservation (Figure 8B–E). However, as vitamin C was associated with the highest levels of vitality and motility of the three antioxidants assessed, it was selected for further testing.

### 3.4. Dose-Dependent Study of the Addition of Water-Soluble Vitamin C Pre- and Post-Cryopreservation

A subsequent analysis demonstrated that the antioxidant impact of vitamin C was clearly dose-dependent (*p* < 0.001) and resulted in antioxidant activity that was statistically different from the antioxidant activity observed following dilution with Quinn’s alone at doses above 0.4 mM (Figure 9A). Analysis of sperm vitality both before and after cryopreservation subsequently demonstrated that this parameter was significantly improved by the addition of vitamin C at a dose of 0.4 mM (Figure 9B,C), while higher doses were inhibitory. Exactly the same profile was observed in the context of DNA damage, which was significantly improved by the addition of 0.4 mM vitamin C following cryopreservation; however, this effect was lost at higher doses (Figure 9D). A similar trend was also observed for sperm motility, with the highest mean value recorded for spermatozoa incubated with 0.4 mM vitamin C; however, this difference was not statistically significant (Figure 9E), again emphasising the resistance of sperm motility to the ameliorating effects of antioxidant supplementation.

## 4. Discussion

The purpose of this study was to determine the impact of cryopreservation reagents and subsequent cryopreservation on the quality of human spermatozoa. For this purpose, we employed a freezing medium that is in widespread clinical use (Quinn’s Advantage Sperm Freeze) and investigated the ways in which this medium and the cryopreservation process impacted the redox status and functionality of human spermatozoa. In the final section of this study, we describe our preliminary attempts to improve the efficacy of the cryopreservation process through the addition of exogenous antioxidants.

Under in vivo conditions, seminal plasma contains a variety of enzymatic antioxidants, including glutathione peroxidase (GPx4), superoxide dismutase (SOD), and catalase, as well as a range of small-molecular-mass free radical scavengers, including vitamin C, uric acid and tyrosine, that remove potentially damaging ROS and balance the sample’s redox potential [32,33,34,35]. Analysis of two forms of antioxidant activity (hydrogen peroxide scavenging and free radical scavenging) in human semen samples before and after the addition of Quinn’s cryoprotection medium revealed that the latter possessed negligible intrinsic antioxidant activity, such that when it was added to human semen samples in a 1:1 ratio, the levels of extracellular antioxidant protection provided to these cells was essentially halved (Figure 1). This finding is important because the distribution and volume of cytoplasmic space available to house antioxidants in this cell type is extremely limited and essentially confined to the midpiece [36]. As a result, spermatozoa are uniquely dependent on the availability of antioxidants in the extracellular space for protection against attack by free radicals. Therefore, it was hypothesized that the significant loss of antioxidant capacity that resulted when semen was diluted 1:1 with Quinn’s medium would render this fluid vulnerable to oxidative stresses generated as a result of cryopreservation [6,10,37,38].

It is common practice to limit semen exposure to cryoprotective agents at room temperature to 8–10 min due to glycerol’s cytotoxic tendencies and the high osmolarity typical of such media [39,40,41,42,43]. Consistent with this practice, we found that the inherent cytotoxicity of Quinn’s cryoprotectant medium precipitated a rapid loss of sperm motility even in the absence of any evidence of oxidative stress. Following cryopreservation, however, clear evidence of oxidative stress was observed. Levels of ROS generation increased, particularly from the mitochondria and particularly after the cells had been thawed and incubated at 37 °C. Thus, whether such cryostored cells are destined for insemination in vivo or in vitro, it is likely that they will be actively generating mitochondrial ROS by the time they make contact with the oocyte. Having antioxidants available to protect frozen-thawed spermatozoa is therefore essential; in this light, the observed reduction in antioxidant protection associated with the addition of Quinn’s medium may be critical. As a consequence of this lack of ambient antioxidant protection, the intracellular generation of mitochondrial ROS resulted in high levels of lipid peroxidation, as evidenced by lipid aldehyde formation (MDA and 4-HNE) and DNA fragmentation (Figure 4, Figure 5 and Figure 6). These lipid aldehydes bound preferentially to the mitochondria of the sperm midpiece (Figure 5), in keeping with the results of previous studies indicating the affinity of these aldehydes for proteins in the mitochondrial electron transport chain, such as succinic acid dehydrogenase [31]. Critically, such powerful adduction of mitochondrial proteins in spermatozoa has been shown to promote ROS generation by these cells in a self-perpetuating chain reaction [31]. Aldehyde adduction of the proteins in the acrosomal region of spermatozoa is also known to compromise the capacity of these cells for sperm-zona recognition, while aldehyde binding to elements of the sperm flagellum, particularly the dynein heavy chain, is thought to be responsible for the close association between lipid peroxidation and impaired sperm motility [31].

Given this set of associations, we examined the ability of several antioxidants to enhance the performance of sperm cryopreservation. The first antioxidant examined was resveratrol. The reagent has been used before to optimise the cryopreservation of human semen and was found to be valuable in some studies [44] but not others [45]. We conducted a detailed dose-response assay with this compound and found that it improved the antioxidant properties of the cryopreservation medium, as anticipated (Figure 7A). However, at doses at which antioxidant activity was significantly elevated (>40 µM), motility was significantly suppressed, indicating the induction of significant reductive stress. Moreover, none of the doses of resveratrol tested improved sperm motility before or after cryopreservation, in keeping with the negative results obtained by Garcez et al. [45].

While resveratrol is an undeniably powerful antioxidant, the hydrophobic nature of the molecule requires a solvent such as ethanol to achieve solubilization in aqueous media. We therefore extended our search for an appropriate antioxidant to include three water-soluble reagents: vitamin C, melatonin, and NAC. All three of these reagents were capable of increasing the levels of antioxidant activity recorded in the semen–cryopreservation mixture such that the levels of antioxidant protection provided to the spermatozoa were not significantly different from those recorded in the initial semen samples (Figure 8A). However, only melatonin and vitamin C were capable of improving the vitality of the spermatozoa prior to preservation (Figure 8B) (when there was no evidence of oxidative stress), and none of these reagents had any significant impact on the vitality or motility of the spermatozoa following cryopreservation (when there was evidence of oxidative stress).

To determine whether this lack of activity might be a function of dose, we conducted a more detailed dose-dependent study with vitamin C. Despite being extremely hydrophilic, this antioxidant is able to enter the sperm interior via the ascorbic acid transporters SVCT1 and SVCT2, both of which are expressed by human spermatozoa [46]. With this set of samples, vitamin C was able to improve sperm vitality and reduce the levels of sperm DNA damage observed following cryopreservation, but, again, while sperm motility was improved, this improvement did not achieve statistical significance (Figure 9). The positive benefits of vitamin C supplementation are generally supported by the literature [47,48,49]. Mangoli et al. [47] found improved sperm motility, vitality, and DNA integrity post-vitrification with the addition of 0.6 mM vitamin C to the semen pre-cryopreservation. Li et al. [49] also found a beneficial impact of vitamin C on the viability of human spermatozoa and on the levels of DNA damage sustained by these cells following cryopreservation. The latter study also found a small positive effect on motility (38.1 ± 7.9 vs. 43.5 ± 10% motile) at 0.3 mM vitamin C, although this effect was lost at a dose (0.6 mM) that had been found to be helpful in vitrification studies [47,49]. An explanation for such mixed responses to different doses of vitamin C can possibly be found in this compound’s pro-oxidant action. As a powerful reducing agent, vitamin C is able to reduce trace amounts of transition metals, such as iron and copper, that can participate in Fenton-like reactions which lead to ROS generation and increased oxidative damage [16,50,51]. The levels of redox-active iron and copper in human semen are variable and in some cases, may be sufficient to promote Fenton chemistry and enhance rather than suppress the induction of oxidative stress [52,53].

In addition, the impact of vitamin C may vary according to any given sample’s vulnerability to reductive stress. It is clear from our dose-dependent studies with both resveratrol and vitamin C that the dose of any antioxidant added to sperm cryopreservation media has to be considered with care. At low doses, some positive effects are observed following antioxidant supplementation; however, as the dose is increased, the cells rapidly go into a state of reductive stress and any therapeutic benefit is reversed (Figure 7 and Figure 9). The positive benefit of antioxidant therapy observed in this study is in keeping with the important role that ROS-induced lipid peroxidation plays in the loss of sperm function and DNA integrity associated with cryopreservation (Figure 4, Figure 5 and Figure 6). The finding of improved DNA integrity, in particular, is in keeping with the results obtained by others with a variety of antioxidants, including ebselen [54], resveratrol [55], vitamin C [55], canthaxanthin [56], curcumin [57], L-carnitine [58], green tea extract [59], catalase [60], Tempol [61], alpha lipoic acid [62], and vitamin E [63]. Such results reinforce the conclusion that DNA damage in cryostored spermatozoa is primarily induced by oxidative stress, rather than by alternative mechanisms such as apoptosis [37]. Motility, on the other hand, has not been as amenable to antioxidant support.

While some studies have demonstrated an improvement in sperm motility following cryopreservation in antioxidant-supplemented media, others have not [54]. Indeed, one study reported a negative relationship between the intrinsic antioxidant activity of human semen and the post-thaw recovery of sperm motility [64]. As we have clearly shown that sperm motility is negatively impacted by ROS generation and lipid peroxidation (Figure 4) and is negatively correlated with DNA damage (Figure 6), such results are surprising. They suggest that there are other factors involved in the suppression of sperm motility following cryopreservation that cannot be ameliorated by antioxidant treatment. One possibility might be indicated by the increased beat cross frequency (BCF) observed following cryopreservation (Figure 2F). BCF provides information about the rate at which flagellar waves are initiated in the spermatozoa; this rate is, in turn, driven by oscillations in intracellular calcium [65]. An increase in BCF therefore suggests an increase in the intracellular calcium content of spermatozoa as a result of cryopreservation, in keeping with the findings of previously published studies [66,67]. While the cryopreservation-induced increase in intracellular calcium remains within physiological limits, it can induce physiological changes, such as an increase in BCF and many of the hallmarks of sperm capacitation [68]. However, when the rise in intracellular calcium exceeds those limits, pathological changes, including a loss of motility, are observed [69]. If the motility loss associated with cryopreservation can be induced by several mechanisms, it would be reasonable to predict that antioxidant supplementation might have variable effectiveness in ameliorating the cryoinjury sustained by any given sample, depending on the influence of other factors like poor calcium homeostasis. On this basis, antioxidant supplementation is only one of the many strategies we must employ if we are to see significant improvements in the cryopreservation of human semen.

### Limitations

Normozoospermic donors were used in the execution of this project in order to establish the basic principles of antioxidant action in the context of cryopreservation. Whether the positive impact of vitamin C would also be observed with unselected patients consulting for male infertility or fertility preservation will have to be established in more extensive clinical trials involving larger numbers of subjects.

## 5. Conclusions

In summary, addition of a commercial cryopreservation medium to human semen has been found to dramatically reduce the levels of antioxidant protection provided to human spermatozoa during the freeze–thaw process. This loss of antioxidant protection is important because cryopreservation is characterised by the induction of oxidative stress, with significant increases in mitochondrial ROS generation and lipid aldehyde formation that are, in turn, negatively correlated with sperm motility, viability, and DNA integrity. Attempts to ameliorate this situation through the addition of antioxidants such as vitamin C improved both sperm vitality and DNA integrity but were less effective in the protection of sperm motility. The use of antioxidants in this context is challenging for two major reasons: (i) the dose-response curve is bell shaped due to the vulnerability of spermatozoa to both oxidative and reductive stress [70], and (ii) complex behaviours such as sperm motility can be modulated by many different factors, some of which will not be responsive to antioxidant supplementation. Future studies in this area will have to recognise the powerful impact of antioxidant dose on sperm function and the parallel operation of additional pathophysiological mechanisms that will also have to be addressed in refining future cryopreservation agents.

## Figures and Tables

**Figure 1 antioxidants-13-00247-f001:**
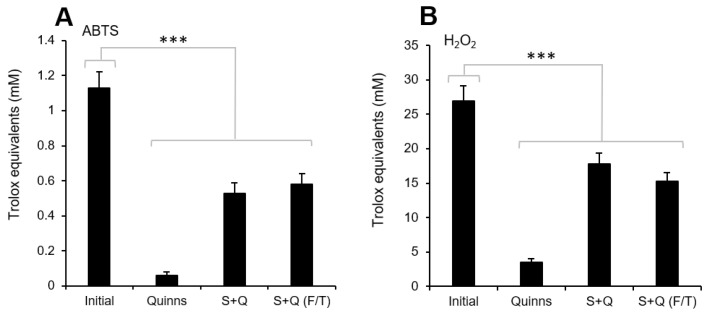
Levels of antioxidant capacity in human semen (Initial) and cryoprotectant (Quinns), as well as in semen diluted with cryoprotectant before (S + Q) and after [S + Q (F/T)] cryopreservation. (**A**) Using an assay to assess ABTS^+●^ radical cation-scavenging activity, little to no antioxidant activity was found in Quinn’s. As a result, the levels of antioxidant protection provided to the spermatozoa were halved following the dilution of semen 1:1 with Quinn’s and remained unchanged by cryopreservation. (**B**) Similarly, little antioxidant activity was found in Quinn’s in terms of its ability to scavenge H_2_O_2_, resulting in a significant loss of antioxidant protection when it was added 1:1 to semen before and after cryopreservation. *** *p* < 0.001; *n* = 15.

**Figure 2 antioxidants-13-00247-f002:**
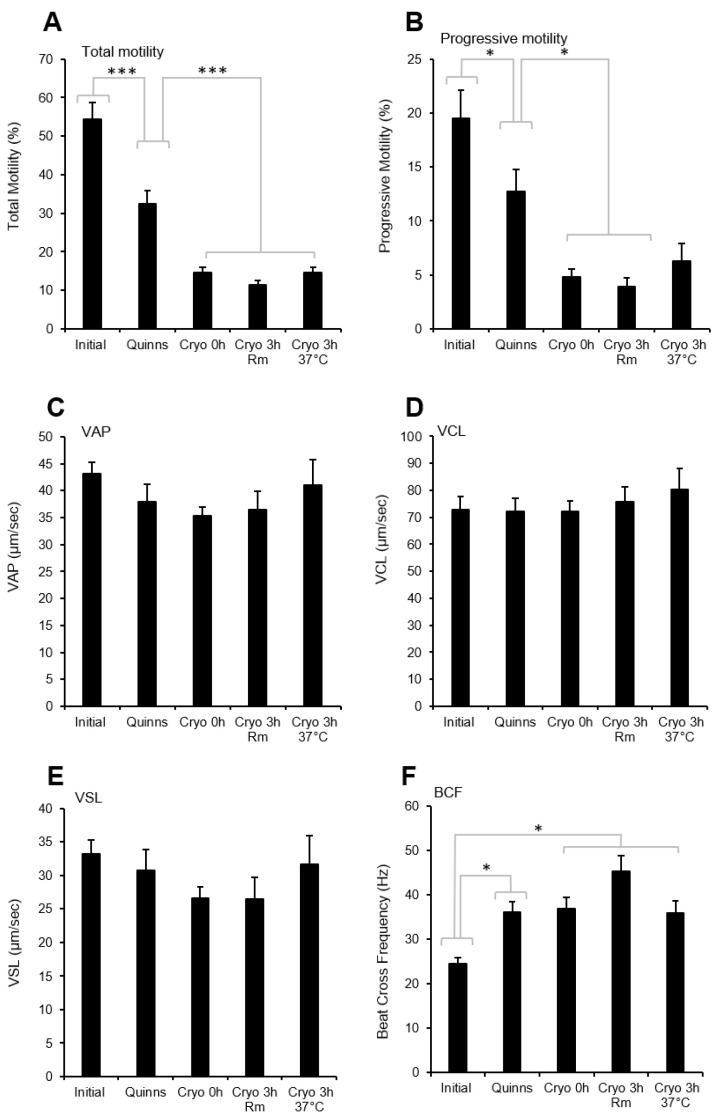
Evidence of cryodamage to human sperm motility following cryoprotectant addition and immediately post-thaw (Cryo 0h), as well as after a three-hour incubation at room temperature (Cryo 3h Rm) or 37 °C (Cryo 3h 37 °C): (**A**) Total motility (**B**) Progressive motility (**C**) Average path velocity (VAP); (**D**) Average curvilinear velocity (VCL); (**E**) Average straight-line velocity (VSL); (**F**) Beat cross frequency.* *p* < 0.05; *** *p* < 0.001; *n* = 15.

**Figure 3 antioxidants-13-00247-f003:**
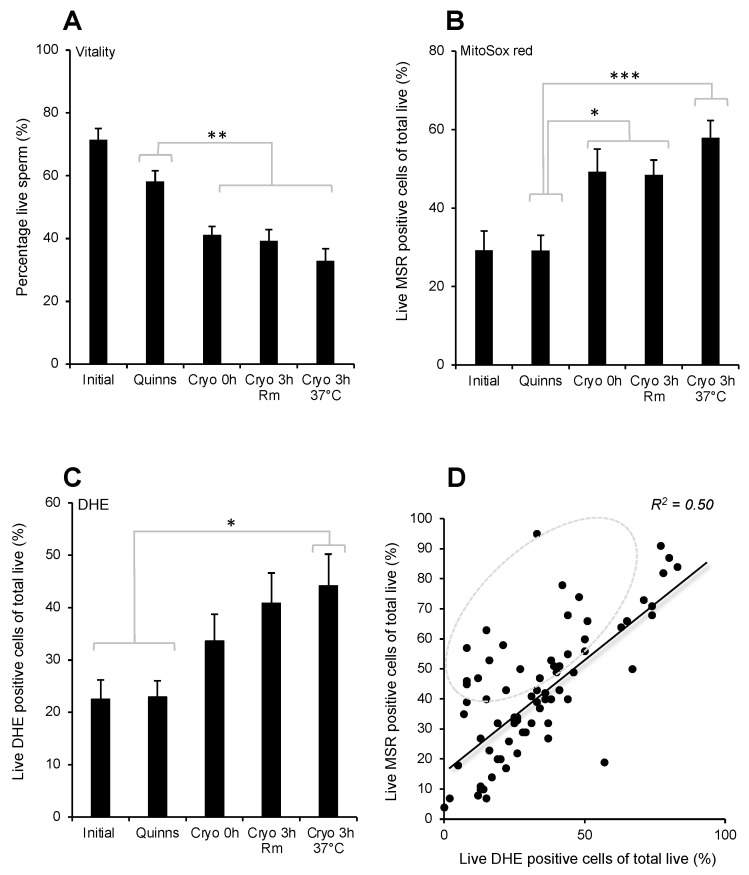
Vitality and ROS generation following cryoprotectant addition (Quinn’s) and immediately post-thaw (Cryo 0 h) as well as after a three-hour incubation at room temperature (Cryo 3 h Rm) or 37 °C (Cryo 3 h 37 °C). (**A**) Vitality (**B**) Mitochondrial ROS generation according to MSR. (**C**) Cellular ROS generation according to DHE. (**D**) Correlation between the levels of mitochondrial and cellular ROS generation in live cells across the entire dataset (*p* < 0.001; R^2^ = 0.5). Some samples were characterised by particularly high mitochondrial ROS generation (circled). * *p* < 0.05; ** *p* < 0.01; *** *p* < 0.001; *n* = 15.

**Figure 4 antioxidants-13-00247-f004:**
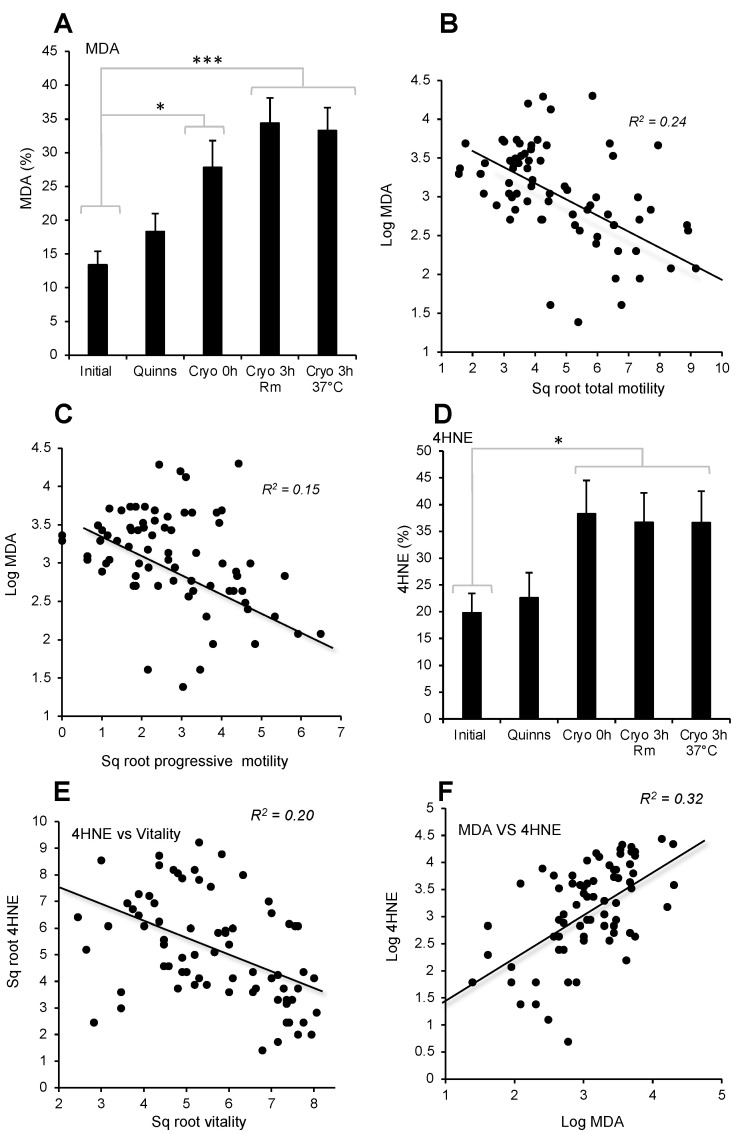
Lipid-peroxidation biomarkers following cryoprotectant addition (Quinns) and immediately post-thaw (Cryo 0 h) as well as after a three-hour incubation at room temperature (Cryo 3 h Rm) or 37 °C (Cryo 3 h 37 °C) (**A**) MDA formation before and after cryopreservation. (**B**) Correlation between MDA and total motility (*p* < 0.001; R^2^ = 0.24). (**C**) Correlation between MDA and progressive motility (*p* < 0.001; R^2^ = 0.15); (**D**) 4-HNE formation before and after cryopreservation. (**E**) Correlation between 4-HNE and vitality (*p* < 0.001; R^2^ = 0.2); (**F**) Correlation between MDA and 4-HNE (*p* < 0.001; R^2^ = 0.32). * *p* < 0.05; *** *p* < 0.001; *n* = 15.

**Figure 5 antioxidants-13-00247-f005:**
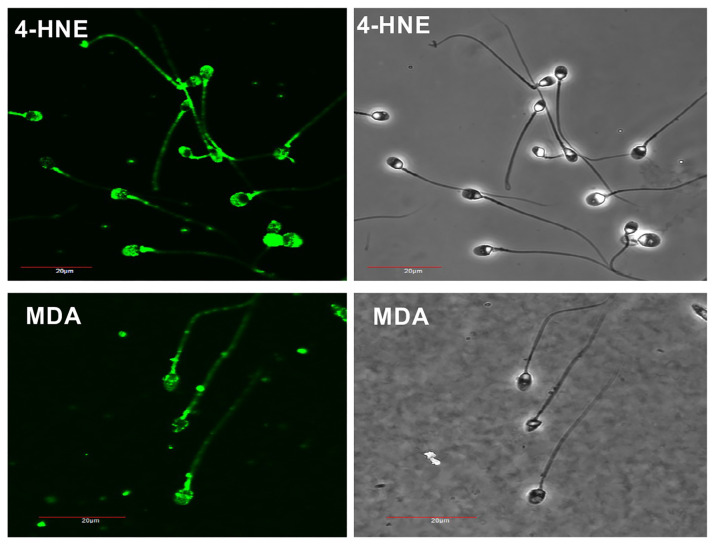
Patterns of lipid aldehyde adduction observed in human spermatozoa. Antibodies against both 4-HNE and MDA revealed the mitochondria of the sperm midpiece to be particularly vulnerable to aldehyde adduction. The acrosome domain was also targeted, particularly by 4HNE, while weak punctate labelling along the length of the flagellum was also observed in some cells.

**Figure 6 antioxidants-13-00247-f006:**
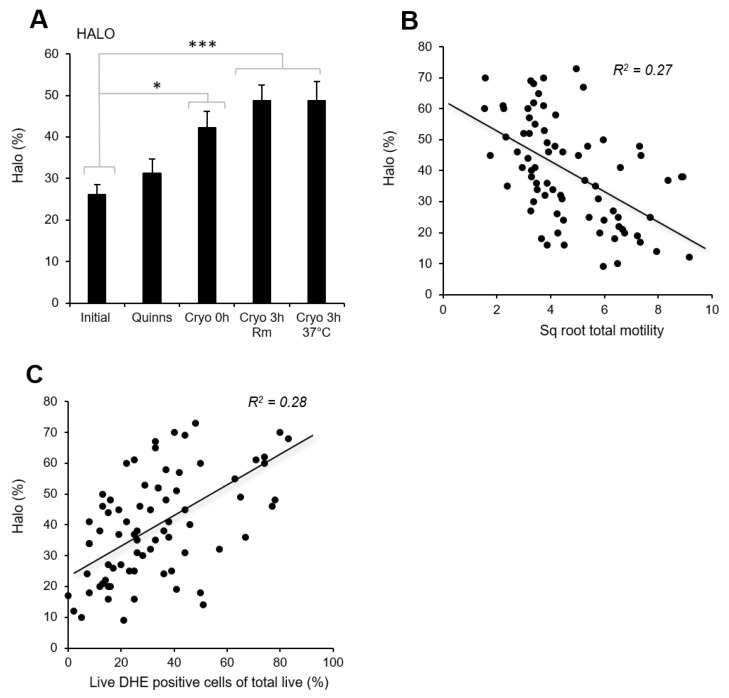
The levels of DNA fragmentation in human spermatozoa subjected to cryoprotectant addition (Quinns) and immediately post-thaw (Cryo 0 h), as well as after a three-hour incubation at room temperature (Cryo 3 h Rm) or at 37 °C (Cryo 3 h 37 °C). (**A**) DNA integrity assessed with the Halo assay. (**B**) Correlation between DNA damage and total cell motility (*p* < 0.001; R^2^ = 0.27); (**C**) Correlation between DNA damage and levels of ROS generation, as measured by DHE (*p* < 0.001; R^2^ = 0.28). * *p* < 0.05; *** *p* < 0.001; *n* = 15.

**Figure 7 antioxidants-13-00247-f007:**
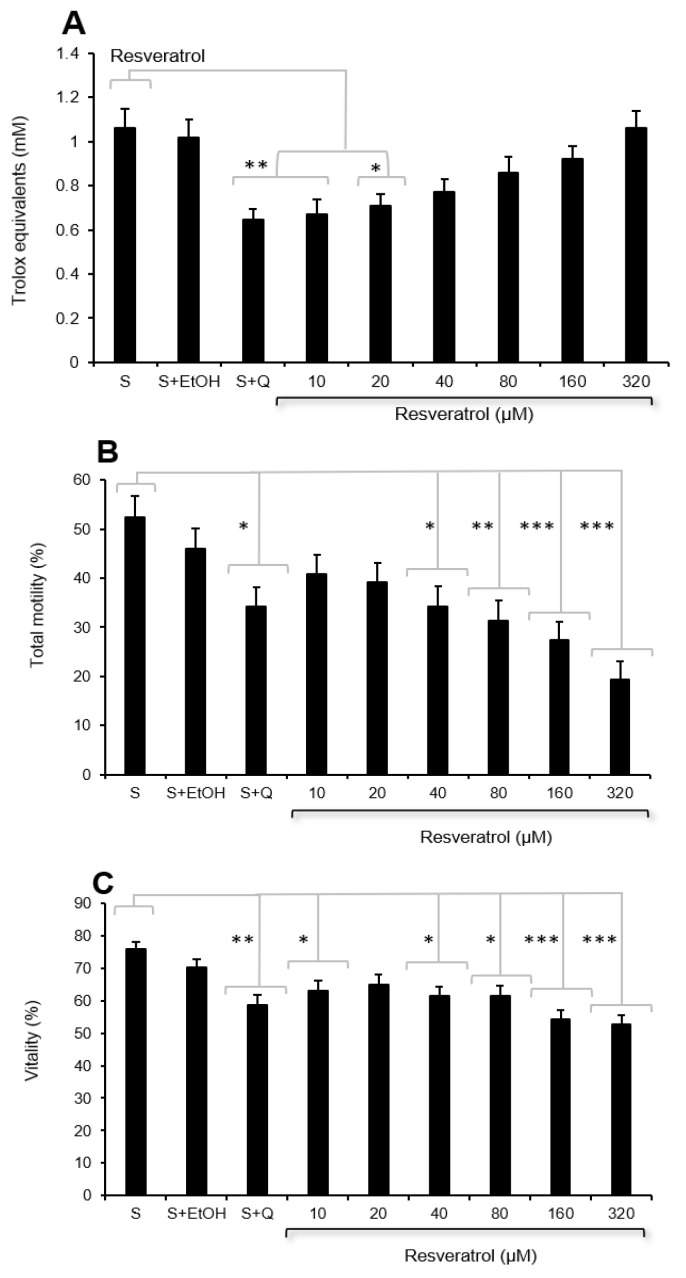
Impact of resveratrol on sperm quality in Quinn’s cryoprotective medium (S + Q) added 1:1 in human semen; ethanol was used to solubilise the resveratrol, necessitating the inclusion of an ethanol control (S + EtOH). (**A**) Changes in antioxidant activity measured with the ABTS^+●^ radical-scavenging assay. (**B**) Changes in sperm motility (**C**) Changes in sperm vitality. * *p* < 0.05; ** *p* < 0.01; *** *p* < 0.001; *n* = 15.

**Figure 8 antioxidants-13-00247-f008:**
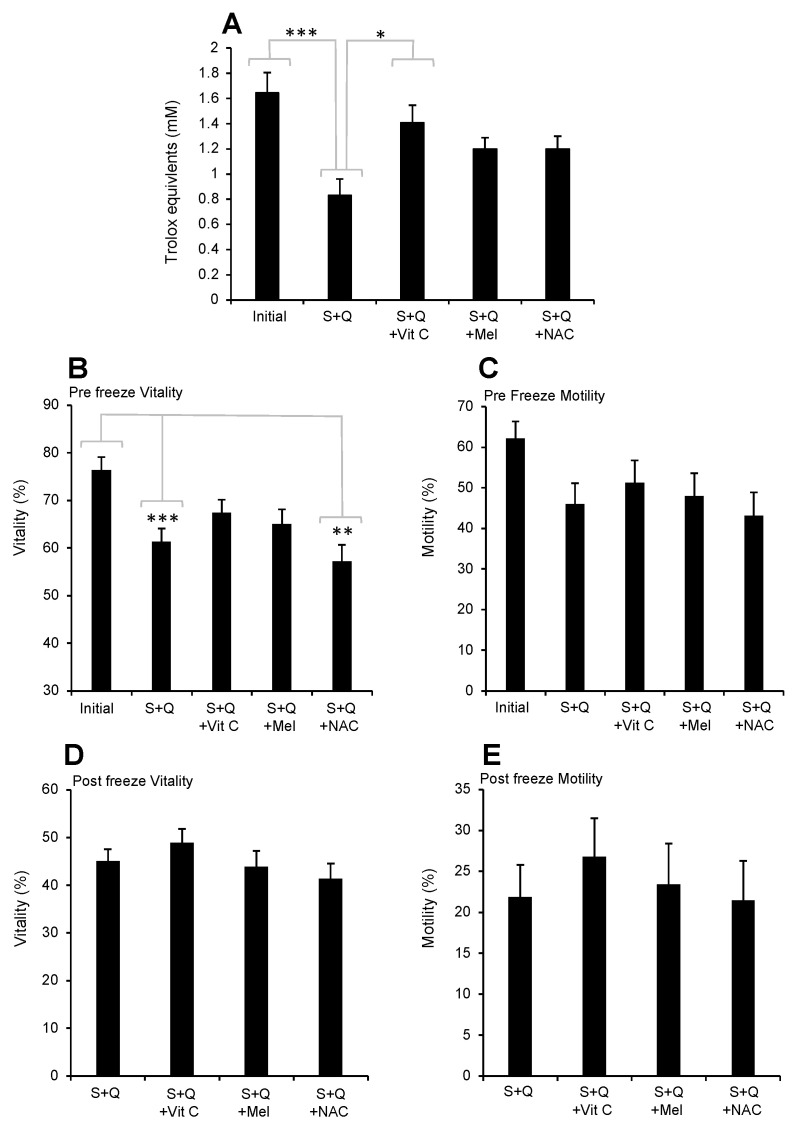
Antioxidant capacity of human semen when Quinn’s cryoprotectant is added 1:1 (S + Q) or supplemented with 0.4 mM vitamin C (S + Q + Vit C), 2 mM melatonin (S+Q + Mel), or 0.2 mM NAC (S + Q + NAC). (**A**) Free radical scavenging activity measured with the ABTS^+●^ radical-scavenging assay. (**B**) Vitality of the spermatozoa prior to cryopreservation. (**C**) Motility of the spermatozoa prior to cryopreservation. (**D**) Vitality of the spermatozoa following cryopreservation. (**E**) Motility of the spermatozoa following cryopreservation. * *p* < 0.05; ** *p* < 0.01; *** *p* < 0.001; *n* = 15.

**Figure 9 antioxidants-13-00247-f009:**
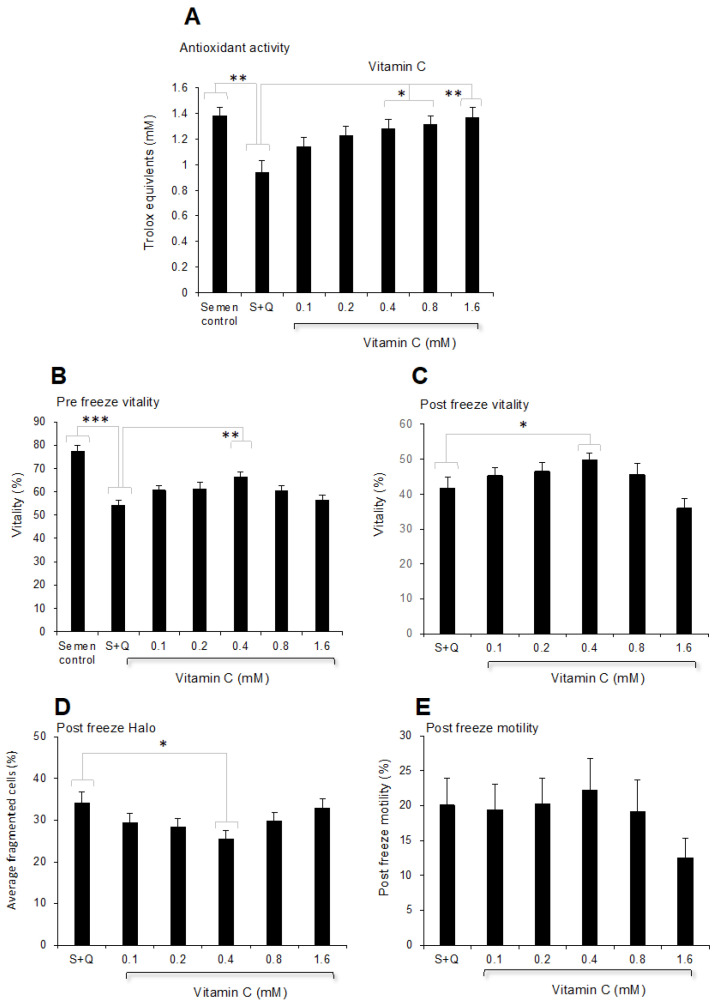
Impact of vitamin C-supplemented cryoprotectant on human semen samples. (**A**) Dose-dependent increase in antioxidant activity, as measured by the ABTS^+●^ radical-scavenging assay. (**B**) Vitality before cryopreservation. (**C**) Vitality following cryopreservation. (**D**) Levels of DNA fragmentation, as measured by the Halo assay following cryopreservation. (**E**) Motility following cryopreservation. * *p* < 0.05; ** *p* < 0.01; *** *p* < 0.001; *n* = 15.

## Data Availability

The data presented in this study are available in the article.

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
