# Peer review of "Addition of Vitamin C Mitigates the Loss of Antioxidant Capacity, Vitality and DNA Integrity in Cryopreserved Human Semen Samples"

_antioxidants, 2024, doi:10.3390/antiox13020247_

Round 1

Reviewer 1 Report

In this paper, Hungerford et al. investigated the changes in antioxidant capacity of human sperm samples subsequent to cryoprotectant addition. 

They also examined the ability of some antioxidant molecules to enhance the performance of sperm cryopreservation. 

The topic of the article is interesting, as the development of an optimized cryoprotective agent should promote high fertilization rates. 

I suggest the following revisions:

1. Title should be changed: it does not include the effects of antioxidant supplementation.

2. The same Authors recently reported that cryopreservation led to the loss of sperm motility and vitality in association with increases in lipid peroxidation and DNA damage (doi: 10.1530/RAF-22-0133). This paper should be introduced in the references.

3. In the Methods section (2.2), authors should describe the characteristics of the 15(?) patients.

4. The limitations of the study should be added at the end of the Discussions.

1. Title should be changed: it does not include the effects of antioxidant supplementation.

2. The same Authors recently reported that cryopreservation led to the loss of sperm motility and vitality in association with increases in lipid peroxidation and DNA damage (doi: 10.1530/RAF-22-0133). This paper should be introduced in the references.

3. In the Methods section (2.2), authors should describe the characteristics of the 15(?) patients.

4. The limitations of the study should be added at the end of the Discussions.

Reviewer 2 Report

No

Comments to the authors:

The authors evaluated the effects of exposure to a commercial cryoprotectant and subsequent cryopreservation on the parameters and redox balance of human sperm. They then explored the protective effects of different antioxidants added to the cryopreservation media. Overall, the idea has clinical application. This manuscript can add valuable information to the existing literature. Therefore, I would like to suggest some minor modifications to improve its quality:

- The title does not include the antioxidant effect of exogenous antioxidants.

- The aim of the study in the abstract is unclear.

- The method of the study is not completely explained in the abstract.

- To avoid overlap between keywords and the title, I would suggest modifying the keywords section.

- In line 46, the authors should add details on the possible mechanisms of sperm sensitivity to cryopreservation. For example, the lipid composition of the spermatozoa membrane plays a significant role in determining the sperm’s ability to withstand freezing and sensitivity to cold.

- In the introduction section, the authors should add information about the antioxidant effects of resveratrol, melatonin, N-acetyl cysteine, and vitamin C on sperm and/or other reproductive cells or tissues based on previous studies.

- In the subsection “Cryopreservation and Thawing of Samples,” the cryoprotectant equilibration, cooling, and thawing procedures are according to the manufacturer’s instructions or previous studies. In the second option, the reference should be cited.

- In lines 113-114, the authors should note the temperature of the cryoprotectant. They also should explain why the sperm were exposed to the cryoprotectant for 8 minutes. The toxicity of the cryoprotectant and post-thaw cell viability is affected by temperature, time, and cryoprotectant concentration. So, “maximum 8 minutes” is confusing. In fact, the cryoprotectant exposure time must be consistent between samples.

- In line 107, I am wondering if the cryoprotectant was removed from the samples or not before the further assessments. After thawing, the cryoprotectant should be removed from the samples to prevent further damages. This procedure should be explained in line 107.

- Previous studies showed that polyvinyl alcohol improves post-thaw motility. I am wondering if the frozen-thawed sperm were treated with polyvinyl alcohol. If so, it should be added to line 119. If not, there should be an explanation for the difference in treatment with PVA between the control and frozen groups.

- In lines 118-119, frozen-thawed sperm cells were analyzed both immediately and following incubation in the dark for 3 h at room temperature (24°C) or 37°C. The authors should note that why they selected these temperatures and the 3 hours incubation time.

- In lines 137-138, I am wondering if the authors conducted dose-response experiments for 2 mM melatonin and 0.2 mM N-acetyl cysteine. If so, please add it to the methodology. If not, references should be cited.

- To make the results section more attractive for readers, I would suggest presenting representative figures for some experiments like fluorescent microscopy and flow cytometry assessment.

- The limitation(s) of the current study should be stated.
